# Are NLP Models Good at Tracing Thoughts: An Overview of Narrative Understanding

**Lixing Zhu**[a*]  **Runcong Zhao**[a*]  **Lin Gui**[a]  **Yulan He**[ab†]

[a]Department of Informatics, King's College London
[b]The Alan Turing Institute, UK
{Lixing.Zhu,Runcong.Zhao,Lin.Gui,Yulan.He}@kcl.ac.uk

## Abstract

Narrative understanding involves capturing the author's cognitive processes, providing insights into their knowledge, intentions, beliefs, and desires. Although large language models (LLMs) excel in generating grammatically coherent text, their ability to comprehend the author's thoughts remains uncertain. This limitation hinders the practical applications of narrative understanding. In this paper, we conduct a comprehensive survey of narrative understanding tasks, thoroughly examining their key features, definitions, taxonomy, associated datasets, training objectives, evaluation metrics, and limitations. Furthermore, we explore the potential of expanding the capabilities of modularized LLMs to address novel narrative understanding tasks. By framing narrative understanding as the retrieval of the author's imaginative cues that outline the narrative structure, our study introduces a fresh perspective on enhancing narrative comprehension.

## 1 Introduction

When reading a narrative, it is common for readers to analyze the author's cognitive processes, including their knowledge, intentions, beliefs, and desires (Castricato et al., 2021; Kosinski, 2023). In general, narrative is a medium for personal experiences (Somasundaran et al., 2018). Although Large Language Models (LLMs) have the capability to generate grammatically coherent texts, their ability to accurately capture the author's thoughts, such as the underlying skeletons or outline prompts devised by the authors themselves (Mahowald et al., 2023), remains questionable. This is supported by cognitive research that bilingual individuals tend to convey more precise thoughts compared to monolingual English speakers (Chee, 2006). The potential deficiency in tracing thoughts within narratives would hinder the practical application of

narrative understanding and, thereby preventing readers from fully understanding the true intention of the authors.

Narrative understanding has been explored through various approaches that aim to recognize thoughts within narratives (Mostafazadeh et al., 2020; Kar et al., 2020; Lee et al., 2021; Sang et al., 2022). Still, these approaches are often fragmented, focusing on diverse tasks scattered across multiple datasets, obfuscating the fundamental elements (e.g., the characters, the events and their relationships) of the narrative structure (Ouyang and McKeown, 2014, 2015; Cutting, 2016). To address this gap, in this paper, we lay the foundation and provide a comprehensive synthesis of the aforementioned narrative understanding tasks. We start with the key features of this genre, followed by a formal definition of narrative understanding. We then present a taxonomy of narrative understanding tasks and their associated datasets, exploring how these datasets are constructed, the training objectives, and the evaluation metrics employed. We proceed to investigate the limitations of existing approaches and provide insights into new frontiers that can be explored by leveraging current modularized LLMs (e.g., GPT with RLHF (Ouyang et al., 2022)), with a particular focus on potential new tasks.

To sum up, our work firstly aligns disparate tasks with the LLM paradigm, and categorizes them based on the choices of context and input-output format. Then it catalogues datasets based on the established taxonomy. Subsequently, it introduces Bayesian prompt selection as an alternative approach to define the task of narrative understanding. Finally, it outlines open research directions.

## 2 Definition of Narrative Understanding

Narrative texts possess distinct characteristics, which are different from other forms of discourse. Elements such as point of view, salient characters,

---

*Equal contribution.
†Corresponding author.

and events, which are associated or arranged in a particular order (Chambers and Jurafsky, 2008; Ouyang and McKeown, 2015; Piper et al., 2021), giving rise to a cohesive story synopsis known as the plots (Hühn et al., 2014). Its scope spans across various genres, including novels, fiction, films, theatre, and more, within the domain of literary theory (Genette, 1988). Although it is unnecessary to endorse a particular narrative theory, some elements are commonly encountered in comprehension. For example, readers have to understand the causation or relationship that goes beyond a timeline and delve into the relationships between the characters (Worth, 2004). From a model-theoretic perspective, narrative understanding can be described as a process through which the audience perceives the narrator's constructed plot or thoughts (Czarniawska, 2004; Castricato et al., 2021).

To this end, we define narrative understanding as the process of reconstructing the writer's creative prompts that sketch the narrative structure (Ouyang and McKeown, 2014; Fan et al., 2019). In line with Brown et al. (2020), we adopt the practice of using the descriptions of NLP tasks as context to accommodate different paradigms. Additionally, we employ the LLaMA (Touvron et al., 2023) taxonomy to dichotomize this data-oriented task as either multiple-choice or free-form text completion. Let $\{x_n, y_n\}_{n=1}^N$ denote the dataset where $x_{1:N}$ are narratives, and $y_{1:N}$ are the annotated sketches, narrative understanding aims to predict $Y$ given $X$ by optimizing $p_\theta(y_{1:N}|x_{1:N}, \text{context})$. Existing literature can be roughly categorized based on the format of $y_n$ and how context is described. To align with the classical NLP taxonomy, we specify context as a single prompt from either Reading Comprehension (Section 2.1), Summarisation (Section 2.2), or Question Answering (Section 2.3). In the rest of this section, the context will be further elaborated and specialized in more narrowly-defined tasks to refine the taxonomy, resulting in the reformatting of $y_n$ accordingly.

## 2.1 Narrative Reading Comprehension

In machine reading comprehension, the context prompt is instantiated as a single prompt of "selecting which option is consistent with the story", and $y_n$ is structured as categorical label(s) that correspond to the available options.

**Narrative Consistency Check** involves determining whether an assertion aligns with the narrative or contradicts it. This task encompasses various scopes, ranging from the entire narrative structure (Ouyang and McKeown, 2014) to discourse structure (Mihaylov and Frank, 2019) and ultimately, the constituents of the narrative, such as agents and events (Piper et al., 2021; Wang et al., 2021).

For example, Granroth-Wilding and Clark (2016) designed a Multiple Choice Narrative Cloze (MCNC) prediction task, where stories are structured as a sequence of events. Each event is represented by a 3-tuple, which comprises the verb lemma, the grammatical relation, and the associated entity. They aimed to predict the subsequent event from a given set of options, framed in the context of story cloze. Furthermore, Chaturvedi et al. (2017) extended this prediction task to encompass the prediction of a story ending based on its existing content. Similarly, the ROC story cloze task (Mostafazadeh et al., 2016), addressed by Cai et al. (2017), involves choosing the most plausible ending. There are various approaches developed for story ending prediction, such as the incorporation of commonsense knowledge (Li et al., 2018b), utilization of skip-thought embeddings (Srinivasan et al., 2018), entity-driven recurrent networks (Henaff et al., 2017; Liu et al., 2018), scene structure (Tian et al., 2020), centrality or salience of events (Zhang et al., 2021), and contextualized narrative event representation (Wilner et al., 2021), respectively. Simple and well-established, the Story Cloze Test does not cover the core aspects of narrative structure, though. Roemmele and Gordon (2018a) introduced an advancements in this task by predicting causally related events in stories using the Choice of Plausible Alternatives (COPA) (Roemmele et al., 2011) dataset. Each instance in the COPA dataset contains three sentences: a *Premise*, *Alternative 1* and *Alternative 2*, with the *Premise* describing an event and the *Alternatives* proposing the plausible cause or effect of the event. Building upon this, Qin et al. (2019) aligned the ROC story cloze and COPA dataset with HellaSwag (Zellers et al., 2019) and introduced the counterfactual narrative reasoning. This task involves re-writing the story to restore narrative consistency. Their proposed TimeTravel dataset features 29,849 counterfactual revisions to initial story endings. Ippolito et al. (2020) further

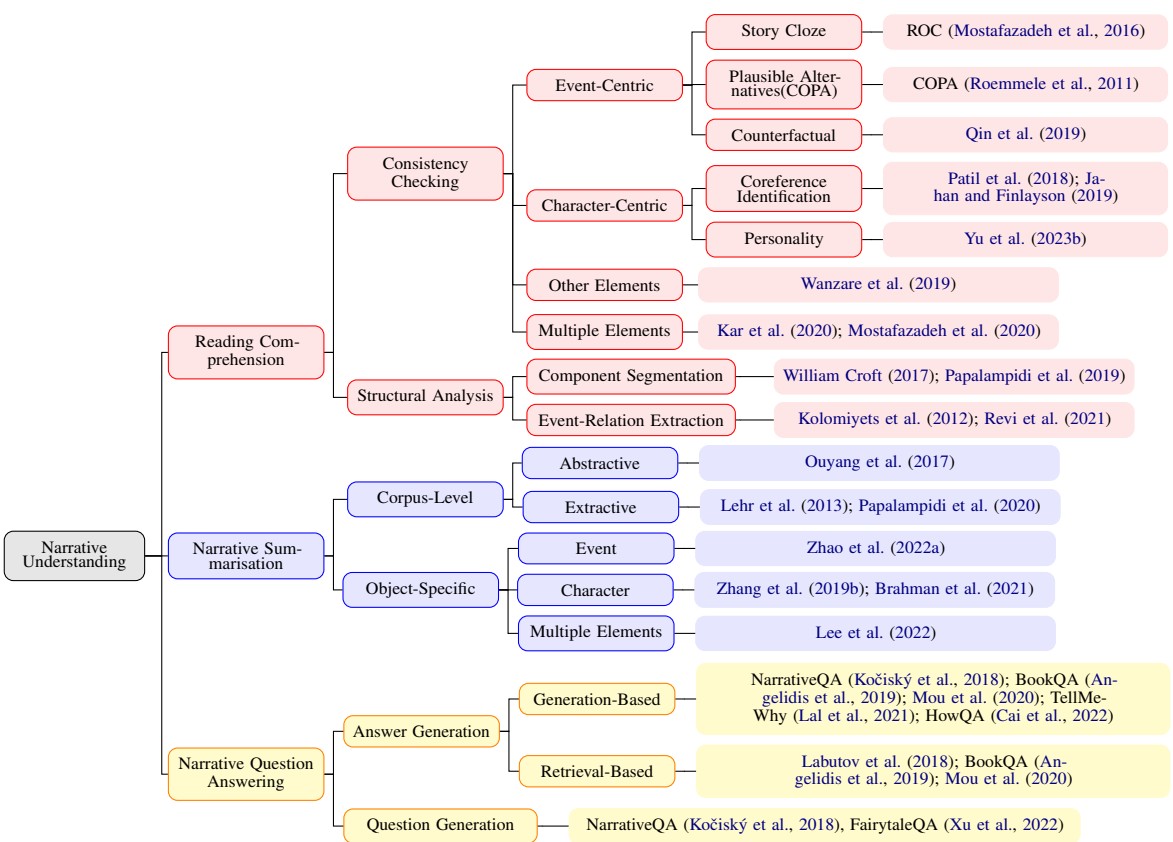

Figure 1: Typology of Narrative Understanding. Some literature sources are repeated since they contain both types of datasets or input-output schemes.

expanded the scope of the story cloze task to the entire narrative and proposed a sentence-level language model for consecutive multiple-choice prediction among candidate sentences on the Toronto Book Corpus (Zhu et al., 2015).

In addition to event prediction, Wanzare et al. (2019) introduced the concept of prototypical events, referred to as scenarios, to incorporate essential commonsense knowledge in narratives. They also introduced a benchmark dataset for scenario detection. Building upon their work, Situated Commonsense Reasoning (Ashida and Sugawara, 2022) aimed at deriving possible scenarios following a given story. Other attributes, such as a movie's success and event salience, have been explored in studies (Kim et al., 2019; Otake et al., 2020; Wilmot and Keller, 2021).

In line with the event consistency, characters, also known as participants, play a crucial role in linking narratives (Patil et al., 2018; Jahan and Finlayson, 2019). Patil et al. (2018) employed Markov Logic Network to encode linguistic knowledge for the identification of aliases of participants in a narrative. They defined participants as entities and framed the task as Named Entity Recog-

nition (NER) and dependency parsing. In contrast, a recent approach introduced the TVShowGuess dataset, which simplified speaker guessing as multiple-choice selection (Sang et al., 2022). However, it is difficult for some narrative-specific NLP tasks, e.g., NER, to determine whether labelling 'a talking cup' as a 'person' is appropriate. To mitigate this, some approaches take a character-centric perspective. Brahman et al. (2021) introduced two new tasks: Character Identification, which assesses the alignment between a character and an unidentified description, and Character Description Generation, which emphasizes generating summaries for individual characters. Other work targeted personality prediction (Yu et al., 2023b), or used the off-the-shelf LLM to perform role extraction (Stammbach et al., 2022). To delve into the psychology of story characters and understand the causal connections between story events and the mental states of characters, Rashkin et al. (2018) introduced a dataset, StoryCommonsense, which contains the annotations of motivations and emotional reactions of story characters.

While much existing work on narrative understanding focuses on specific aspects, Kar et al.

(2020) considered the multifaceted features of narratives and created a multi-label dataset from both plot synopses and movie reviews. Chaturvedi et al. (2018) identified narrative similarity in terms of plot events, and resemblances between characters and their social relationships. Mostafazadeh et al. (2020) built the GeneraLized and COntextualized Story Explanations (GLUCOSE) dataset from children's stories and focused on explaining implicit causes and effects within narratives. It includes events, locations, possessions, and other attributes of the curated claims within the stories.

**Structural Analysis of Events: Plot and Storyline Extraction** The narrative structure encompasses a sequence of events that shape a story and define the roles of its characters (Hearst, 1997; Cutting, 2016). Unlike narrative consistency checking, which focuses on elementary consistency, this task involves two key objectives. First, it aims to extract a clear and coherent timeline that underlies the narrative's progression. Second, it aims to sequence the relationship of key factors to construct the plot of the narrative (Kolomiyets et al., 2012).

In the first line of approaches, Ouyang and McKeown (2015) considered significant shifts, referred to as Turning Points, in a narrative. These turning points represent the reportable events in the story. Li et al. (2018a) divided a typical story into five parts: *Orientation*, *Complicating Actions*, *Most Reportable Event*, *Resolution* and *Aftermath*, which are annotated in chronological order, capturing the temporal progression of the story. The temporal relationships within the narrative can be extracted and structured into a database of temporal events (Yao and Huang, 2018). In a similar vein, Papalampidi et al. (2019) strived to identify turning points by segmenting screenplays into thematic units, such as setup and complications. Anantharama et al. (2022) developed a pipeline approach that involves event triplet extraction and clustering to reconstruct a time series of narrative clusters based on identified topics.

Based on the types of event relations such as temporal, causal, or nested, storylines can be organised as timelines (Ansah et al., 2019; yang Hsu et al., 2021), hierarchical trees (Zhu and Oates, 2012), or directed graphs (Norambuena and Mitra, 2021; Yan and Tang, 2023). Current approaches often construct storylines using stated timestamps (Ansah et al., 2019; Revi et al., 2021). However, challenges arise in narratives where time details may be vague

or absent. To address this issue, William Croft (2017) proposed to decompose the storyline by considering individual temporal subevents for each participant in a clausal event, which interact causally. Bamman et al. (2020) focused on resolving coreference in English fiction and presented the LitBank to resolve the long-distance within-document mentions. Building upon this work, Yu et al. (2023a) released a corpus of fiction and Wikipedia text to facilitate anaphoric reference discovery. Yan et al. (2019) introduced a more complex structure called Functional Schema, which utilizes language models, to reflect how storytelling patterns make up the narrative. Mikhalkova et al. (2020) introduces the Text World Theory (Werth, 1999; Wang et al., 2016) to regulate the structured annotations of narratives. This annotation scheme, profiling the world in narrative, is expanded in (Levi et al., 2022) by adding new narrative elements. Situated reasoning datasets, such as *Moral Stories* (Emelin et al., 2021), target the branching developments in narrative plots, specifically focusing on if-else scenarios. Tools have also been created for annotating the semantic relations among the text segments (Raring et al., 2022).

In addition to the aforementioned approaches for plot or storyline construction, several joint models have been developed to simultaneously uncover key elements and predict their connections. A notable work is PlotMachines (Rashkin et al., 2020) which involves an outline extraction method for automatic constructing the outline-story dataset. In another study, Lee et al. (2021) employed Graph Convolutional Networks (GCN) to predict entities and links on the StoryCommonsense and DesireDB datasets (Rahimtoroghi et al., 2017).

## 2.2 Narrative Summarisation

Narrative summarisation, often referred to as Story Retelling (Lehr et al., 2013), can be specified by restricting $y_n$ to be a paraphrase that captures the essense of the original literature. Similar to other summarisation tasks, narrative summarisation can be extractive or abstractive, depending on whether the paraphrase is text snippets directly extracted from the story or is generated from input text.

Early tasks, such as Automated Narrative Retelling Assessment (Lehr et al., 2013), primarily focused on recapitulation story elements in the form of a tagging task. Subsequently, Narrative Summarisation Corpora (Ouyang et al., 2017; Papalampidi et al., 2020) were developed to facilitate

more comprehensive understandings of narratives. The former is designed for abstractive summarization, while the latter is intended for informative sentence retrieval, taking into account the inherent narrative structure. IDN-Sum (Revi et al., 2020) provides a unique view of summarisation within the context of Interactive Digital Narrative (IDN) games. Recent work has proposed benchmarks that require machines to capture specific narrative elements, e.g., synoptic character descriptions (Zhang et al., 2019b) and story rewriting anchored in six dimensions (Lee et al., 2022). In the same vein, Goyal et al. (2022) collected span-level annotations based on discourse structure to evaluate the coherence of summaries for long documents. Brahman et al. (2021) presented a character-centric view by introducing two new tasks: Character Identification and Character Description Generation. More recently, a benchmark called NarraSum (Zhao et al., 2022a) has been developed for large-scale narrative summarisation, encompassing salient events and characters, albeit without explicit framing.

## 2.3 Narrative Question Answering

Answering implicit, ambiguous, or causality questions from long narratives with diverse writing styles across different genres requires a deep level of understanding (Kalbaliyev and Sirts, 2022). From the task perspective, Narrative QA can be categorized based on either the format of $y_n$, or the task prompt that is referred to as context. The format of $y_n$ could be categorical, where an answer is provided as a span specified by starting-ending positions. Alternatively, it can be free-form text that is generated. The context could be specified as answer selection/generation or question generation.

Numerous research works have focused on providing accurate answers to curated questions, with a specific focus on event frames (Tozzo et al., 2018), or questions related to external commonsense knowledge (Labutov et al., 2018). They all fall into the category of retrieval-based QA, where relevant information is selected from narratives. In contrast, the NarrativeQA (Kočiský et al., 2018) dataset took a different approach by instructing the annotators to ask questions and express answers in their own words after reading an entire long document, such as a book or a movie script. This resulted in high-quality questions designed by human annotators, and human-generated answers. The dataset further provided supporting snippets from human-written abstractive summaries and the original story.

To effectively handle long context, Tay et al. (2019) introduced a Pointer-Generator framework to sample useful excerpts for training, and chose between extraction and generation for answering. Meanwhile, the BookQA (Angelidis et al., 2019) approach targeted *Who* questions in the NarrativeQA corpus by leveraging BERT to locate relevant content. Likewise, Mou et al. (2020) proposed a two-step approach which consists of evidence retrieval to build a collection of paragraphs and a question-answering step to produce an answer given the collection. Mou et al. (2021) surveyed open-domain QA techniques and provided the Ranker-Reader solution, which improves upon the work of Mou et al. (2020) with a newer ranker and reader model.

Unlike pipeline approaches, Mihaylov and Frank (2019) converted the free-text answers from NarrativeQA into text-span answers and used the span answers as labels for training and prediction. Other attempts have been made to adapt NarrativeQA for extractive QA. For example, Frermann (2019) modified the dataset into an extractive QA format suitable for passage retrieval and answer span prediction. On the other hand, the TellMeWhy (Lal et al., 2021) dataset combined the commonsense knowledge and the characters' motivations in short narratives when designing the questions, presenting a new challenge for answering why-questions in narratives. Kalbaliyev and Sirts (2022) reviewed the WhyQA challenges, and Cai et al. (2022) collected how-to questions from *Wikihow* articles to build the HowQA dataset, which serves as a testbed for a Retriever-Generator model.

Generating meaningful questions is an important aspect of human intelligence that holds great educational potential for enhancing children's comprehension and stimulating their interest (Yao et al., 2022; Zhang et al., 2022a). Early work in this area focused on generating questions of multiple choice word cloze from children's Books, targeting named entities, nouns, verbs and prepositions (Hill et al., 2016). Other studies designed commonsense-related questions from narratives in simulated world (Labutov et al., 2018). To enhance children's learning experiences, the FairytaleQA (Xu et al., 2022) dataset was created for question generation (QG) tasks, covering seven types of narrative elements or relations. The dataset was used in the work of (Yao et al., 2022), which employed a

pipeline approach comprising heuristics-based answer generation, BART-based question generation, and DistilBERT-based candidate ranking. Zhao et al. (2022b) experimented with a question generation model, which incorporated a summarisation module, using the same FairytaleQA dataset. Additionally, the StoryBuddy (Zhang et al., 2022a) system provided an interactive interface for parents to participate in the process of generating question-answer pairs, serving an educational purpose.

A major taxonomical concern in tasks or datasets employing similar input-output schemes, particularly NarrativeQA (Kočiský et al., 2018) with the objective of answering questions relating to narrative consistency, is the potential overlap between finely-detailed tasks as defined in our taxonomy. For example, depending on the instructions and prompts given, the scope of QA can encompass more specific tasks, including narrative summarization (Who is Charlie?), and narrative generation (What would happen after the Princess marries the Prince?). Despite some similarities, the majority of instruction templates are indistinguishable (Sanh et al., 2022). Therefore, they are considered as equivalent tasks.

## 3   Dataset

We have conducted a review of datasets (Table 1-3 in Appendix) that are either designed for, or applicable to, tasks related to narrative understanding. There are three major concerns:

**Data Source**   Due to copyright restrictions, most datasets focus on public domain works. This limitation has rendered valuable resources, such as the Toronto Book Corpus (Zhu et al., 2015), unavailable. To overcome this challenge, diverse sources have been explored, such as plot descriptions from movies and TV episodes (Frermann et al., 2018; Zhao et al., 2022a). However, there remains a need for datasets that cover specialized knowledge bases for specific worldviews in narratives, such as magical worlds, post-apocalyptic wastelands, and futuristic settings. One potential data source that could be utilized for this purpose is TVTropes [1], which provides extensive descriptions of character traits and actions.

**Data Annotation**   The majority of existing datasets contain short stories, consisting of only a few sentences, which limits their usefulness for

understanding complex narratives found in books and novels. Due to high annotation costs, there is a lack of sufficiently annotated datasets for these types of materials (Zhu et al., 2015; Bandy and Vincent, 2021). Existing work (Frermann et al., 2018; Chaudhury et al., 2020; Kryscinski et al., 2022) employed available summaries and diverse meta-information from books, movies, and TV episodes to generate sizable, high-quality datasets. Additionally, efficient data collection strategies, such as game design (Yu et al., 2023a), character-actor linking for movies (Zhang et al., 2019b), and leveraging online reading notes (Yu et al., 2023b), can be explored to facilitate the creation of datasets.

**Data Reuse**   Despite the availability of high-quality annotated datasets for various narrative understanding tasks, there is limited reuse of these datasets in the field. Researchers often face challenges in finding suitable data for their specific tasks, which leads to the creatation of their own new and costly datasets. Some chose to build a large, general dataset (Zhu et al., 2015; Mostafazadeh et al., 2020), while others chose to gradually annotate the same corpus over time (Frermann et al., 2018). Some made use of platforms such HuggingFace for data sharing (Huang et al., 2019), or provided dedicated interfaces for public access (Koupaee and Wang, 2018). To facilitate data reuse and address the challenges associated with finding relevant data, the establishment of an online repository could prove beneficial.

## 4   Evaluation Methods

For classification tasks, such as multiple-choice QA and next sentence prediction, accuracy, precision, recall, and the F1 score serve as the most suitable evaluation metrics. Here we mainly discuss evaluation metrics for free-form or generative tasks, crucial yet challenging for narrative understanding. Traditional metrics based on N-gram overlap like BLEU (Papineni et al., 2002), ROUGE (Lin and Hovy, 2003), METEOR (Lavie and Agarwal, 2007), CIDER (Vedantam et al., 2015), WMD (Kusner et al., 2015), and their variants, such as NIST (Doddington, 2002), BLEUS (Lin and Och, 2004), SacreBLEU (Post, 2018) remain popular due to their reproducibility. However, their correlation with human judgment is weak for certain tasks (Novikova et al., 2017; Gatt and Krahmer, 2018).

Alternative content overlap measures have been proposed, such as PEAK (Yang et al., 2016), which

---

[1] http://tvtropes.org

compares weighted Summarization Content Units, and SPICE (Anderson et al., 2016), which evaluates overlap by parsing candidate and reference texts into scene graphs representing objects and relations. The advent of BERT introduced a new approach to evaluation, based on contextualized embeddings, as seen in metrics such as BERTScore (Zhang et al., 2019a), MoverScore (Zhao et al., 2019), and BARTScore (Yuan et al., 2021). Task-specific evaluations using BERT-based classifiers, such as FactCC (Kryscinski et al., 2020) for summary consistency and QAGS (Wang et al., 2020) to determine if answers are sourced from the document, have been explored.

Recently, LLMs have also been employed for evaluation tasks. For instance, CANarEx (Anantharama et al., 2022) assesses time-series cluster recovery from GPT-3 generated synthetic data. UniEval (Zhong et al., 2022) uses a pretrained T5 model for evaluation score computation. GPTScore (Fu et al., 2023) leverages zero-shot capabilities of ChatGPT for text scoring, while G-Eval (Liu et al., 2023) applies LLMs within a chain-of-thoughts framework and form-filling paradigm for output quality assessment.

To facilitate tracking the research progress, we list the performance of current SoTA models on representative benchmarks in Table 4 in the Appendix. These benchmarks have been categorized into specific tasks based on the instruction templates outlined in Section 2. It is worth noting that methodological development or result analysis is not the focus of this survey. Sections 2.1-2.3 are intended to serve as timelines depicting the evolution of different research directions and task comparisons. Further, Section 3 offers a review of the diverse datasets and their respective formats.

## 5 Applications

**Narrative Assessment**   The "good at language -> good at thought" fallacy (Mahowald et al., 2023) has spurred the application of narrative understanding to the automatic assessment of student essays. Studies on automatic story evaluation (Chen et al., 2022; Chhun et al., 2022) reveal that the referenced metrics, e.g., BLEU and ROUGE scores, deviate from human preferences such as aesthetic and intrigue. This calls for the identification of narrative elements and their relations. For example, Somasundaran et al. (2016) builds a graph of discourse proximity of essay concepts to predict the essay quality w.r.t. the development of ideas and exemplification. Somasundaran et al. (2018) annotates multiple dimensions of narrative quality, such as narrative development and narrative organization, to combat the scarcity of scored essays. Such narrativity is also evaluated in (Steg et al., 2022) with a focus on detecting the cognitive aspects, i.e., suspense, curiosity, and surprise. Other sub-tasks, such as comment generation, are studied in (Lehr et al., 2013; Zhang et al., 2022b).

**Story Infilling**   As mentioned in Section 2.1, the story cloze task selects a more coherent ending based on the story context. The story infilling completes a story in a similar way that sequences of words are removed from the text to create blanks for a replacement (Ippolito et al., 2019). Mori et al. (2020) makes a step forward in detecting the missing or flawed part of a story. The proposed method predicts the positions and provides alternative wordings, which serves as a writing assistant. It is worth mentioning that narrative generation intersects with this task with the key difference that story infilling aims to comprehend the narratives and generate minor parts to complete the story. Wider applications of this task are auxiliary writing systems such as an educational question designer (Zhang et al., 2022a) and a creative writing supporter (Roemmele and Gordon, 2018b).

**Narrative Understanding vs. Narrative Generation**   The main distinction we make between narrative understanding and generation is that the latter aims to produce longer sequences conditioned on prototypical story snippets. An epitome is story generation conditioned upon prompts (Fan et al., 2019), where the prompts draft the action plan and reserve the placeholders for generative models to complete. Unlike Story Infilling, the out-of-distribution narrative elements and their relations, e.g., plot structures (Goldfarb-Tarrant et al., 2020), novel plots (Ammanabrolu et al., 2019), creative interactions (DeLucia et al., 2021), and interesting endings (Gupta et al., 2019), are to be generated, which poses the main challenge in story generation (Goldfarb-Tarrant et al., 2019). Other literature focuses on the significant enrichment of details to a brief story skeleton (Zhai et al., 2020). Latent discrete plans illustrated by thematic keywords are posited (Peng et al., 2018), and latent variable models are leveraged to steer the generation (Jhamtani and Berg-Kirkpatrick, 2020). Commonsense reasoning also needs to be pondered for storytelling

everyday scenarios (Mao et al., 2019; Xu et al., 2020). Evaluation criteria are adjusted accordingly to measure the aesthetic merit and correlate with human preferences (Akoury et al., 2020; Chhun et al., 2022). In this sense, narrative generation is the opposite task of narrative understanding with a key emphasis on generating intriguing plots and rich details based on a small corpus. Despite being the opposite in the model-theoretic view, narrative generation needs to comply with certain restrictions, e.g., pragmatics, coreference consistency (Clark et al., 2018), long-range cohesion (Zhai et al., 2019; Goldfarb-Tarrant et al., 2020), and adherence to genres (Alabdulkarim et al., 2021), to name a few.

## 6 Challenges and Future Directions

**Prompt Tuning and Author Prompt Recapitulation** In lieu of the prompt-driven nature of LLMs that elicits tasks with task descriptions or few-shot examples rather than task-specific fine-tuning (Brown et al., 2020; Sanh et al., 2022; Touvron et al., 2023; Tay et al., 2023; Anil et al., 2023), we propose a unified approach for narrative understanding. This approach involves a single supervised training process, denoted as $p_\theta(y_{1:N}|x_{1:N}, \text{context})$, where context represents a prompt of task description or few-shot examples, and $y_{1:N}$ denotes predictable annotations produced by crowd-sourced platforms such as Amazon Mechanical Turk (Mostafazadeh et al., 2016; Papalampidi et al., 2020; Lal et al., 2021). While the framework is universally applicable, the practice of relying on annotators to infer authors' thoughts or construct skeleton prompts is inefficient, inaccurate and unreliable (Mahowald et al., 2023), Direct consultation with the authors is often impractical, particularly for posthumous masterworks. As suggested in the model-theoretic view (Castricato et al., 2021), a gap needs to be closed between the narrator and the audience to overcome the reader's uncertainty and other environmental limitations.

In this regard, we envisage a Bayesian perspective (Lyle et al., 2020) for the recovery of the author's thoughts. Let $p_\phi(\text{narrative}|\text{sketch}, \text{task})$ denote the oracle model that simulates the author's composition process, where the author fills in the conceived skeleton out of their own initiative (where sketch is the skeleton prompts and task is the task description). The recapitulation of the author's prompt can then be expressed as $\text{argmax}_{\text{sketch}} \, p_\phi(\text{narrative}|\text{sketch})$. Previous ef-

forts have formulated the objective as predicting the narrative elements or the narrative structure (as discussed in Section 2), given the intractability of the likelihood of generating the narrative, $p_\phi(\text{narrative}|\text{sketch}, \text{task})$. It is also viable, however, to probe and optimize sketch directly, considering the ability of LLMs to compute the error in close proximity. Hence, $p_\phi(\text{narrative}|\text{sketch})$ could be derived as the marginal likelihood $\Sigma_{\text{task}} p_\phi(\text{narrative}, \text{task}|\text{sketch})$, and its expectation form $\mathbb{E}_{p(\text{task}|\text{sketch})} p_\phi(\text{narrative}|\text{sketch}, \text{task})$ can be approximated by sampling an appropriate task. The dependence between task and sketch aligns well with the composition practice of choosing a suitable genre to match the author's creative intent. In more complex cases where the narrator is known to have employed particular narratology techniques (e.g., Flashback (Han et al., 2022)), task and sketch can be parameterized by generative models (e.g., LM-driven prompt engineers (Zhou et al., 2023)), which can be tuned by a prompt base through gradient descent optimization.

**Interactive Narrative** LLMs, with their ability to carry out numerous language processing tasks effectively in a zero-shot manner (Shen et al., 2023), are paving the way towards a future of immersive and interactive narrative environments. These environments could resemble the dynamic storylines experienced by individuals, as depicted in the TV series "Westworld".

*Agent* Recent studies (Park et al., 2023; Auto-GPT, 2023) have shown promising results in using a database to store an agent's experiences and thought processes, effectively serving as a personality repository. By retrieving relevant memories from this database and incorporating them into prompts, LLMs can be guided to predict behaviors, thereby producing human-like responses. However, extracting comprehensive character-centric memory from narratives, encompassing aspects such as "Who", "When", "Where", "Action", "Feeling", "Causal relation", "Outcome", and "Prediction" (Xu et al., 2022), remain largely unexplored. Current studies primarily focus on simple corpora and there is ample room for further investigation in this area.

*Environment* Creating immersive and interactive environments for users and agents presents several challenges, primarily due to three key factors: **(1) Environment Extraction**. Character locations and environments, often vaguely defined unless crucial

to the plot, have to be clarified. Most works rely on pre-built sandbox environments (Côté et al., 2018; Hausknecht et al., 2020; Park et al., 2023) to address this issue. However, challenges remain in extracting and representing the environment accurately. **(2) Environment Generation**. Interactive narratives aim to provide users with greater freedom, but this poses the challenge of automatically generating reasonable and coherent details within the narrative's world. It is crucial to maintain consistency and engagement in storytelling, despite varying user inputs and directions. **(3) Environment Update**. Agents' text commands may change the state of the world, requiring accurate and cost-effective updates. Current systems update environment states using predefined rules (Côté et al., 2018; Wang et al., 2022). However, using LLMs to derive and generate narrative environments challenges the use of predefined rules, making efficient and large-scale environment updates a future research direction.

**Open World Knowledge** Incorporating external knowledge and commonsense has been a long-standing challenge in both dataset construction and model design (Zellers et al., 2019; Wanzare et al., 2019; Mikhalkova et al., 2020; Ashida and Sugawara, 2022). Efforts to address this challenge have been made over the years, with the emergence of LLMs providing a source potential of commonsense knowledge (Bosselut et al., 2019; Petroni et al., 2019). Notably, the Text World Theory (TWT) (Werth, 1999; Gavins, 2007) has been leveraged to provide world knowledge relevant to everyday life, which is simulated through natural language descriptions (Labutov et al., 2018). Similarly, in (Mikhalkova et al., 2020), a text world framework is established, in which the world-building elements (e.g., characters, time and space) are annotated to enhance readers' perception.

The text world entails nuanced knowledge derived from the interplay of various elements. However, such supervision provided by the static world is somewhat limited, as the reward is implicit and the model needs to extrapolate from the annotations to exploit the world knowledge. In contrast, the open world (Raistrick et al., 2023) presents an ideal source of supervisor, where the model can be rewarded with incentives derived from world mechanisms (Assran et al., 2023) that synergistically complements the audience model with the everyday commonsense. To this end, the RLHF (Reinforce-

ment Learning with Human Feedback) (Ouyang et al., 2022) strategy could be applied to the open-world system, which enables the iterative training of the narrative understanding process.

## 7 Conclusion

In this paper, we have systematically examined the emerging field of narrative understanding, cataloguing the approaches and highlighting the unique challenges it poses along with the potential solutions that have emerged. We have emphasized the crucial role of LLMs in advancing narrative understanding. Our intention is for this survey to serve as a thorough guide for researchers navigating this intricate domain, drawing attention to both the commonalities and unique aspects of narrative understanding in relation to other NLP research paradigms. We aspire to bridge the gap between existing works and potential avenues for further development, thus inspiring meaningful and innovative progress in this fascinating field.

## Limitations

This paper provides an overview of narrative understanding tasks, drawing inspiration from computational narratology (Matthews et al., 2003) and exploring potential new directions. However, it does not delve into broader concepts of cognitive computational theory, such as the theory of mind (Happé, 1994), the philosophy of reading (Mathies, 2020) and pedagogics (Nicolopoulou and Richner, 2007). Therefore, this survey does not incorporate cognitive-theoretic insights into the underlying mechanisms that contribute to the success of models. Another major limitation is the lack of discussion of methodological improvements, as the focus of the research progression is primarily centered around the tasks. Additionally, this survey does not explore narrative generation tasks.

## Acknowledgments

We would like to thank the anonymous reviewers for their insightful comments and helpful suggestions. This work was funded by the UK Engineering and Physical Sciences Research Council (grant no. EP/T017112/1, EP/T017112/2, EP/V048597/1). YH is supported by a Turing AI Fellowship funded by the UK Research and Innovation (grant no. EP/V020579/1, EP/V020579/2).

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

**Setting For Datasets Related To Narrative Understanding Tasks**

| Dataset | Reference | Task | Evaluation |
|---|---|---|---|
| Toronto Book Corpus | Zhu et al. (2015) | Narrative Understanding | - |
| BookCorpusOpen | Bandy and Vincent (2021) | Narrative Understanding | - |
| ELTeC | Odebrecht et al. (2021) | Narrative Understanding | - |
| GLUCOSE | Mostafazadeh et al. (2020) | Reading Comprehension | BLEU scores + Human |
| ROCStories | Mostafazadeh et al. (2016) | Consistency Checking: Event-Centric | Story Cloze Test |
| ROCStories Winter 2017 | Mostafazadeh et al. (2017) | Consistency Checking: Event-Centric | Story Cloze Test |
| Possible Stories | Ashida and Sugawara (2022) | Consistency Checking: Event-Centric | Accuracy |
| COPA | Roemmele et al. (2011) | Consistency Checking: Plausible Alternatives | Accuracy |
| TIMETRAVEL | Qin et al. (2019) | Counterfactual | Similarity Metrics + Human[1] |
| HellaSwag | Zellers et al. (2019) | Counterfactual | Accuracy |
| StoryCommonsense | Rashkin et al. (2018) | Consistency Checking: Character-Centric | F-score + Explanation score[2] |
| TVShowGuess | Sang et al. (2022) | Consistency Checking: Character-Centric | Accuracy |
| PERSONET | Yu et al. (2023b) | Consistency Checking: Character-Centric | Accuracy |
| LitBank | Bamman et al. (2020) | Coreference Identification | F-score |
| Phrase Detectives | Yu et al. (2023a) | Coreference Identification | Accuracy |
| - | Wanzare et al. (2019) | Consistency Checking: Other Elements | F-score |
| SNaC | Goyal et al. (2022) | Consistency Checking: Multiple Elements | F-score |
| - | Pustejovsky and Stubbs (2011) | Structural Analysis | Accuracy |
| InScript | Modi et al. (2016) | Structural Analysis | - |
| Hippocorpus | Sap et al. (2020) | Structural Analysis | Narrative Flow + Event contains |
| ESC v0.9 | Caselli and Vossen (2021) | Structural Analysis | F-score |
| DesireDB | Rahimtoroghi et al. (2017) | Event-Relation Extraction | F-score |
| Moral Stories | Emelin et al. (2021) | Event-Relation Extraction | F-score + Similarity Metrics[3] |
| CSI | Frermann et al. (2018) | Corpus-Level Summarisation | F-score |
| Shmoop | Chaudhury et al. (2020) | Corpus-Level Summarisation | Accuracy |
| NovelChapter | Ladhak et al. (2020) | Corpus-Level Summarisation | Similarity Metrics |
| BookSum | Kryscinski et al. (2022) | Corpus-Level Summarisation | ROUGE-n, BERTScore, SummaQA |
| NARRASUM | Zhao et al. (2022a) | Corpus-Level Summarisation | ROUGE-n, SummaC |
| IDN-Sum | Revi et al. (2020) | Corpus-Level Summarisation | ROUGE-1 + F-score |
| ABLIT | Roemmele et al. (2023) | Corpus-Level Summarisation | ROUGE-1 + F-score[4] |
| CMU Movie Summary | Bamman et al. (2013) | Character-Centric Summarisation | Variation of information + Purity score |
| - | Zhang et al. (2019b) | Character-Centric Summarisation | Recall@K |
| LiSCU | Brahman et al. (2021) | Character-Centric Summarisation | Accuracy |
| BookTest | Bajgar et al. (2017) | Story Cloze | Accuracy |
| MCTest | Richardson et al. (2013) | Answer Generation | Accuracy |
| Children's Book Test | Hill et al. (2016) | Answer Generation | Accuracy |
| MovieQA | Tapaswi et al. (2016) | Answer Generation | Accuracy |
| WikiHow | Koupaee and Wang (2018) | Answer Generation + Summarisation | METEOR |
| MCScript2.0 | Ostermann et al. (2019) | Answer Generation | Accuracy |
| Cosmos QA | Huang et al. (2019) | Answer Generation | Accuracy |
| NarrativeQA | Kočiský et al. (2018) | Narrative Question Answering | Similarity Metrics |
| TellMeWhy | Lal et al. (2021) | Narrative Question Answering | Similarity Metrics |
| FairytaleQA | Xu et al. (2022) | Narrative Question Answering | ROUGE-L F1 score |

Table 1: Settings for datasets related to narrative understanding tasks, including references, the tasks the dataset was created for, and the evaluation methods. Below are some supplementary information that cannot fit in the table: [1] Similarity Metrics (e.g. BLUE-4, ROUGE-L, BERT, BERT-FT, Word Mover's Similarity, Sentence + Word Mover's Similarity), complemented by human evaluation using Likert scale scores; [2] F-score for category labels, Vector average and extrema score for annotation explanations; [3] Accuracy and F1 score for classification, as well as similarity metrics for generation tasks; [4] ROUGE-1 precision score between spans and F-score for sentence labels.

**Domain Information On Datasets Related To Narrative Understanding Tasks**

| Dataset | Domain | Dataset Size | Average Text Length | Language |
|---|---|---|---|---|
| Toronto Book Corpus | Romance, Historical, Adventure, etc. | 11,038 books | ~6,704 sentences | English |
| BookCorpusOpen | Romance, Historical, Adventure, etc. | 17,868 books | - | English |
| ELTeC | - | 1,250 novels | - | 8 Languages[1] |
| GLUCOSE | Commonsense stories (ROCStories) | 4,881 stories | 5 sentences | English |
| ROCStories | Commonsense Stories | 49,255 stories | 5 sentences | English |
| ROCStories Winter 2017 | Commonsense Stories | 98,159 stories | 5 sentences | English |
| Possible Stories | Short story with multiple endings | 1,313 passages | 46.3 tokens | English |
| COPA | Choice Of Plausible Alternatives | 1K questions | 1 sentence | English |
| TIMETRAVEL | Commonsense stories (ROCStories) | 29,849 story rewritings | 5 sentences | English |
| HellaSwag | Commonsense stories (SWAG) | 70K passage | 1 sentence | English |
| StoryCommonsense | Commonsense stories (ROCStories) | 15K stories | 5 sentences | English |
| TVShowGuess | Scripts of TV series | 318 characters | 137,568 tokens | English |
| PERSONET | Novel | 33 books | 11,876 sentences | English, Chinese |
| LitBank | Fiction | 100 fictions | 2,105.3 tokens | English |
| Phrase Detectives | Fiction and Wikipedia | 805 documents | 1,712.4 tokens | English |
| Wanzare et al. (2019) | Blog (Spinn3r) | 504 stories | 35.74 sentences | English |
| SNaC | LLMs generated book/movie summaries | 150 books | 41 sentences | English |
| Pustejovsky and Stubbs (2011) | - | 183 articles | - | English |
| InScript | Commonsense stories (given scenarios) | 910 stories | 12.4 sentences | English |
| Hippocorpus | Stories of imaged/recalled events | 6,854 Stories | 17.6 sentences | English |
| ESC v0.9 | ECB+ corpus[2] | 258 documents | - | English |
| DesireDB | Blog (Spinn3r) | 3,680 instances | - | English |
| Moral Stories | Social Norms, Morality/Ethics | 12K stories | - | English |
| CSI | Crime Drama | 39 episodes (59 cases) | 689 sentences per case | English |
| Shmoop | Novels, plays, short stories[3] | 231 stories | 112,080 tokens | English |
| NovelChapter | Novel | 4,383 chapters | 5,165 words | English |
| BookSum | Plays, short stories, novels (Gutenberg) | 405 books | 112,885.15 tokens | English |
| NARRASUM | Plot descriptions of Movie/TV episodes | 122K narratives | 786 tokens | English |
| IDN-Sum | Narrative game scripts | 8 IDN episodes | 3250 sentences | English |
| ABLIT | Novels (Gutenberg) | 868 chapters | 154.1 sentences | English |
| CMU Movie Summary | Movie plot summaries | 42,306 movies | 176 words [4] | English |
| Zhang et al. (2019b) | Romance, Werewolf, etc. (Wattpad) | 1,036,965 stories | 15,600 words | English |
| LiSCU | Educational stories | 1,220 books | 1431.2 tokens [5] | English |
| BookTest | Books (Gutenberg) | 14,140,82 questions | 522 tokens | English |
| MCTest | Books (Gutenberg) | 500 stories + 2,000 questions | 212 words [6] | English |
| Children's Book Test | Books (Gutenberg) | 108 books + 687,343 questions | 462.7 /30.7 words | English |
| MovieQA | Movie scripts | 14,944 questions | 9.3 words | English |
| WikiHow | HowWiki website | 230,843 articles | 579.8 tokens | English |
| MCScript2.0 | Short stories around everyday scenarios | 3,487 texts + 19,821 questions | 164.4 / 8.2 tokens | English |
| Cosmos QA | Paragraph + Questions [7] | 35,600 (paragraphs + questions) | 69.4 /10.3 tokens | English |
| NarrativeQA | Books (Gutenberg), movie scripts | 1572 documents + 46,765 questions | 61,472 / 9.8 tokens | English |
| TellMeWhy | Commonsense stories (ROCStories) | 9,636 stories + 30,519 questions | 5 sentences | English |
| FairytaleQA | Classic fairytale stories | 278 stories + 10,580 questions | 1401.3 / 3.3 tokens | English |

Table 2: Domain information on datasets related to narrative understanding tasks, including data domain, dataset size, average text length (per narrative or character), and the language used. Below are some supplementary information that cannot fit in the table: [1] 8 Europoean Language including Czech, German, English, French, Hungarian, Polish, Portuguese and Slovenian; [2] ECB+ corpus focuses on calamity events, such as shooting and accidents; [3] Short story sources include Shmoop website and Gutenberg; [4] The median length is 176 words since no average text length is provided; [5] The length pertains to book summaries; [6] Text length ranges from 150-300 words; [7] Questions relate to causes, effects, facts, and counterfactuals.

**Annotation Information On Datasets Related To Narrative Understanding Tasks**

| Dataset | Total Number of Annotations | Annotation Type | Annotation Procedure |
|---|---|---|---|
| Toronto Book Corpus | - | - | - |
| BookCorpusOpen | - | - | - |
| ELTeC | - | - | - |
| GLUCOSE | ~670K | Commonsense Causal Knowledge | Human-Crowdsourced (MTurk) |
| ROCStories | - | Causal + Temporal Span | Human-Crowdsourced (MTurk) |
| ROCStories Winter 2017 | - | Causal + Temporal Span | Human-Crowdsourced (MTurk) |
| Possible Stories | 8,885 ending + 4,533 questions | Alternative Ending + Causal Question | Human-Crowdsourced (MTurk) |
| COPA | 2 Alternatives for each question | the more Plausible Alternatives | Human |
| TIMETRAVEL | 81,407 counterfactual branch | Counterfactual Rewritings | Human-Crowdsourced (MTurk) |
| HellaSwag | 70K Answers[1] | Counterfactual Reasoning | Machine-generated |
| StoryCommonsense | 55,747 w/motiv + 104,930 w/emot | Motivations + Emotional Reactions | Human |
| TVShowGuess | 12,413 scene | Character Facts | Human (2 experts) |
| PERSONET | 140,268[2] | Personalities Traits | Automatic collection + Human |
| LitBank | 29,103 mentions | Anaphoric Reference + Entity Category | Human (3 experts) |
| Phrase Detectives | 282,558 mentions | Anaphoric Reference | Human-Crowdsourced[3] |
| Wanzare et al. (2019) | 10,754 sentences[4] | Scenarios + Segmentation | Human(4 student assistants) |
| SNaC | 9.6K Span | Coherence Error Span + Type | Human[5] |
| Pustejovsky and Stubbs (2011) | - | Temporal Span | Human (3 students) |
| InScript | 62,062[6] | Script Structure | Human-Crowdsourced (MTurk) |
| Hippocorpus | - | Human Recalled Events | Human-Crowdsourced (MTurk) |
| ESC v0.9 | 9169 relations[7] | Event Relation + Temporal Span | Human (2 students) |
| DesireDB | 3,680 | Desire Expressions[8] | Human-Crowdsourced (MTurk) |
| Moral Stories | 24K action + 48K consequence | Story Segment + Sentence Categories | Human-Crowdsourced (MTurk) |
| CSI | Story Segment + Sentence Categories | Factual/Structural Metadata[9] | Human (3 students) |
| Shmoop | 7,234 summaries for 7,234 chapters | Segmentation + chapter-level summaries | Automatic collection[10] |
| NovelChapter | 8,088 chapter/summary pairs | Chapter-level summaries | Automatic collection + Human written[11] |
| BookSum | 405 summaries | Paragraph, chapter, book-level summaries | Automatic alignment + Human inspection |
| NARRASUM | 122K summaries | Book-level summaries | Automatic alignment + Human inspection |
| IDN-Sum | 10k summaries for 10k documents | Interactive narratives summaries | Automatic collection[12] |
| ABLIT | 868 | Paragraph-level abridged texts | Automatic alignment + Human written[13] |
| CMU Movie Summary | 29,802 characters[14] | Character metadata | Automatic matching |
| Zhang et al. (2019b) | 18,100 characters | Character Metadata + Tropes | Automatic extraction + Human[15] |
| LiSCU | 9499[16] | Character Description | Automatic creation + Human evaluation[17] |
| BookTest | 141,408,250 options | Cloze-form | Automatic creation |
| MCTest | 8000[18] | Multiple-choice | Human-Crowdsourced (MTurk) |
| Children's Book Test | 10 choices for each question | Multiple-choice | Automatic creation |
| MovieQA | 74,720 answers | Multiple-choice | Human |
| WikiHow | 230,843 summaries | Subtopics + Free-form Answer | Automatic collection |
| MCScript2.0 | 2 choices for each question | Answer Generation + Multiple-choice | Human-Crowdsourced (MTurk) |
| Cosmos QA | 4 choices for each question | Multiple-choice | Human-Crowdsourced (MTurk) |
| NarrativeQA | 46,765 answers | Free-form Answer | Human-Crowdsourced (MTurk) |
| TellMeWhy | 3 answers for each question | Free-form Answer | Human-Crowdsourced (MTurk) |
| FairytaleQA | 10,580 | Answer + Ground-truth Question Pairs | Human-5 postgraduate students |

Table 3: Annotation information on datasets related to narrative understanding tasks, including the total number of annotations, annotation type, and annotation procedure (expert vs. crowdsourced, human vs. automatic). Below are some supplementary information that cannot fit in the table: [1] Adversarial wrong answers for each passage; [2] 140,268 traits are derived from 110,114 notes, which were automatically collected from reading apps. These notes were written by the app's users; [3] The data was sourced via a game-with-a-purpose approach; [4] Annotators labelled a total of 504 documents, which comprised 10,754 sentences. A label for a scenario could be assigned from one of the 200 predefined scenarios or marked as "None" for sentences that didn't fit any scenario; [5] Both expert evaluators (3 experts) and human crowdsourcing through MTurk were used for annotation; [6] Stories were annotated across 10 distinct scenarios. Verbs and noun phrases were labelled with event and participant types, respectively. The text also includes coreference annotations. [7] The dataset includes 6,904 temporal relations and 2,265 explanatory relations.; [8] It has gold standard labels for identifying statements of desire, spans of evidence supporting the fulfillment of the desire, and annotations indicating whether the stated desire is fulfilled based on the narrative context; [9] References to the mentioned perpetrator and relation to previous cases; [10] Paired summaries and narrative texts sourced from websites; [11] Summaries and chapter pairs were automatically collected from online study guides, written by experts; [12] Paired summaries and narrative texts sourced from websites; [13] The dataset has an automatically aligned abridged version, which is written by a single human author.; [14] Characters are matched to actors with a public date of birth; [15] Annotations are collected through questionnaires to 100 authors; [16] 1708 literature summaries and 9499 character descriptions; [17] 3 judges were asked to evaluate the quality of the description, focusing on fact coverage and task difficulty; [18] There are 4 questions associated with each story, and each question offers 4 answer choices.

| Task | Representative Dataset | Best Model | Result |
|------|----------------------|-----------|--------|
| StoryCloze | StoryCloze (Mostafazadeh et al., 2016) | FLAN 137B zero-shot (Wei et al., 2022) | 93.4 Accuracy |
| CounterfactualReasoning | Hellaswag (Zellers et al., 2019) | GPT-4 (OpenAI, 2023) | 95.3 Accuracy |
| NarrativeSummarization | BookSum (Kryscinski et al., 2022) | BART-LS (Xiong et al., 2022) | 38.5 Rouge-1 |
| NarrativeQA | NarrativeQA (Kočiský et al., 2018) | Masque (Nishida et al., 2019) | 59.87 Rouge-L |
| NarrativeQA | Children's Book Test (Hill et al., 2016) | NSE (Dhingra et al., 2017) | 71.9 Accuracy |

Table 4: Performance of the most recent models on representative datasets. The results are extracted from their respective papers.