# OpenReview forum: "Are NLP Models Good at Tracing Thoughts: An Overview of Narrative Understanding"
_EMNLP/2023/Conference — EMNLP 2023 Findings_

### Official Review · Reviewer_ZCLn · 2023-08-02

**Typos Grammar Style And Presentation Improvements:** Line 305
**Soundness:** 3

**Excitement:**

3: Ambivalent: It has merits (e.g., it reports state-of-the-art results, the idea is nice), but there are key weaknesses (e.g., it describes incremental work), and it can significantly benefit from another round of revision. However, I won't object to accepting it if my co-reviewers champion it.

**Paper Topic And Main Contributions:**

The paper surveys the narrative understanding task along the lines of reading comprehension, summarization and question answering. The authors postulate that narrative understanding is the task of finding the author's latent thoughts that generated the story. The work suggests that we can estimate these latent thoughts with task descriptions and prompts.

**Questions For The Authors:**

Line 34-37: "This is supported ... English speaker".
How does this result from cognitive research support the support the previous statement about LLMs lacking the ability to capture the author's thoughts?

**Reasons To Accept:**

The main strength of this paper is the discussion surrounding the future directions for narrative understanding tasks, especially the proposal of a Bayesian estimation for the author's thoughts. Additionally, this paper can serve as a good resource for researchers to find an index of narrative understanding datasets.

**Reasons To Reject:**

Being a survey paper, the authors could have strengthened their contribution with a more fine-grained categorization of the tasks and datasets. For example, dataset size, narrative text length, annotation procedure (expert vs crowdsourced, human vs automatic), domain, period, author count, language, evaluation, etc.

A task-specific example for the Bayesian estimation of the author's thoughts is necessary. The corresponding discussion was interesting but difficult to understand without being able to trace to a particular task.

**Reproducibility:**

N/A: Doesn't apply, since the paper does not include empirical results.

**Reviewer Confidence:**

4: Quite sure. I tried to check the important points carefully. It's unlikely, though conceivable, that I missed something that should affect my ratings.

---

> ### Author Rebuttal · Authors · 2023-08-29
>
> We appreciate your review! In response to the weaknesses you’ve pointed out:
> > 1. Being a survey paper, the authors could have strengthened their contribution with a more fine-grained categorization of the tasks and datasets. For example, dataset size, narrative text length, annotation procedure (expert vs crowdsourced, human vs automatic), domain, period, author count, language, evaluation, etc.
>
> The original table for the dataset can be found in Appendix Table A1. Based on the suggestions, we have added more detailed information about those datasets, which is shown in the two tables below. The first table provides information and settings for each dataset, including references, the tasks the dataset was created for, the language used, the data domain, and the evaluation methods. The second table includes statistical and annotation information about each dataset, such as dataset size, average text length (per narrative or character), total number of annotations, annotation type, and annotation procedure (expert vs. crowdsourced, human vs. automatic).
>
> |Dataset | Reference | Task | Language | Domain | Evaluation |
> | :------------: | :-----------: | :-----------: | :-----------: | :-----------: | :-----------: |
> | Toronto Book Corpus |  Zhu et al., 2015 | Narrative Understanding | English | Romance, Historical, Adventure, etc. | - |
> | BookCorpusOpen |  Bandy and Vincent, 2021 | Narrative Understanding | English | Romance, Historical, Adventure, etc. | - |
> | ELTeC |  Odebrecht et al., 2021 | Narrative Understanding | 8 European Languages $^{\mathrm{[1]}}$ | - | - |
> | GLUCOSE |  Mostafazadeh et al., 2020 | Reading Comprehension | English | Commonsense stories (ROCStories) | BLEU scores + Human Evaluation |
> | ROCStories |  Mostafazadeh et al., 2016 | Consistency Checking: Event-Centric | English | Commonsense stories | Story Cloze Test  |
> | ROCStories Winter 2017 |  Mostafazadeh et al., 2017 | Consistency Checking: Event-Centric | English | Commonsense stories | Story Cloze Test  |
> | Possible Stories | Ashida and Sugawara, 2022 | Consistency Checking: Event-Centric | English |Short story with multiple endings | Accuracy |
> | COPA | Roemmele et al., 2011 | | English | Choice Of Plausible Alternatives | Accuracy |
> | TIMETRAVEL | Qin et al., 2019  | Counterfactual | English | Commonsense stories (ROCStories) | Similarity Metrics $^{\mathrm{[2]}}$ + Human Evaluation (Likert scale scores) |
> | HellaSwag | Zellers et al., 2019  | Counterfactual | English | Commonsense stories (SWAG) | Accuracy |
> | StoryCommonsense | Rashkin et al., 2018 | Consistency Checking: Character-Centric | English | Commonsense stories (ROCStories)  | F1 score ( for category labels) + Vector average and extrema score (for annotation explanations) |
> | TVShowGuess | Sang et al., 2022 | Consistency Checking: Character-Centric | English | scripts of TV series | Accuracy |
> | PERSONET | Yu et al., 2023b | Consistency Checking: Character-Centric | English, Chinese | Novel| Accuracy |
> | LitBank | Bamman et al., 2020 | Coreference Identification | English | Fiction | F-score |
> | Phrase Detectives | Yu et al., 2023a | Coreference Identification | English | Fiction and Wikipedia | Accuracy |
> | - | Wanzare et al., 2019 | Consistency Checking: Other Elements | English | Blog (Spinn3r) | F-score |
> | SNaC | Goyal et al., 2022 | Consistency Checking: Multiple Elements | English | LLMs generated book/movie summaries | F-score  |
> | - | Pustejovsky and Stubbs, 2011 | Structural Analysis | English | - | Accuracy |
> | InScript | Modi et al., 2016 | Structural Analysis | English | Commonsense stories under given scenarios | - |
> | Hippocorpus | Sap et al., 2020 | Structural Analysis | English | Stories of imaged/recalled events | Narrative Flow + Event contains |
> | ESC v0.9 | Caselli and Vossen, 2021 | Structural Analysis | English | ECB+ corpus concerning calamity events (shooting, accidents, etc.) | F-score |
> | DesireDB | Rahimtoroghi et al., 2017 | Event-Relation Extraction | English | Blog (Spinn3r) | F-score |
> | Moral Stories | Modi et al., 2016 | Event-Relation Extraction | English | Social Norms, Morality/Ethics | Classification (Accuracy, F1 score) + Generation (Similarity Metrics) |
> | CSI | Frermann et al., 2018 | Corpus-Level Summarisation | English | Crime Drama | F-score |
> | Shmoop | Chaudhury et al., 2020 | Corpus-Level Summarisation | English | Novels, plays and short stories (Shmoop website & Gutenberg) | Accuracy |
> | BookSum | Kryściński et al., 2022 | Corpus-Level Summarisation | English | Plays, short stories, and novels (Gutenberg) | ROUGE-n, BERTScore, SummaQA |
> | NARRASUM | Zhao et al., 2022a | Corpus-Level Summarisation | English | Plot descriptions of Movie/TV episodes |  ROUGE-n, SummaC |
> | IDN-Sum | Revi et al., 2020 | Corpus-Level Summarisation | English | Narrative game scripts | ROUGE-1 F1 |
> | ABLIT | Roemmele et al., 2023 | Corpus-Level Summarisation | English | Novel (Gutenberg) | ROUGE-1 precision score (between spans) + F-score (of sentence labels) |
> | CMU Movie Summary | Bamman et al., 2013 | Character-Centric Summarisation | English | Movie plot summaries | Variation of information + Purity score |
> | - | Zhang et al., 2019b | Character-Centric Summarisation | English | Romance, Werewolf, etc. (Wattpad) | Recall@K |
> | LiSCU | Brahman et al., 2021 | Character-Centric Summarisation | English | Educational stories | Accuracy|
> | BookTest | Bajgar et al., 2017 | Story Cloze | English | Books (Gutenberg) | Accuracy |
> | MCTest | Richardson et al., 2013 | Answer Generation | English | Short fictional stories| Accuracy|
> | Children’s Book Test  | Hill et al., 2016 | Answer Generation | English| Books (Gutenberg) | Accuracy |
> | MovieQA | Tapaswi et al., 2016 | Answer Generation | English | Movie scripts | Accuracy |
> | WikiHow | Koupaee and Wang, 2018 | Answer Generation & Summarisation | English | HowWiki website | METEOR |
> | MCScript2.0 | Ostermann et al., 2019 | Answer Generation | English| Short stories around everyday scenarios | Accuracy |
> | Cosmos QA | Huang et al., 2019 | Answer Generation | English| Paragraph & questions relates to causes, effects, facts, counterfactuals | Accuracy |
> | NarrativeQA | Kočiský, et al., 2018 | Narrative Question Answering | English| Books (Gutenberg), movie scripts | Similarity Metrics |
> | TellMeWhy | Lal et al., 2021 | Narrative Question Answering | English| Commonsense stories (ROCStories) |Similarity Metrics |
> | FairytaleQA | Xu et al., 2022 | Narrative Question Answering | English| Classic fairytale stories | ROUGE-L F1 score |
>
> [1] Czech, German, English, French, Hungarian, Polish, Portuguese and Slovenian
>
> [2] BLUE-4, ROUGE-L, BERT, BERT-FT, Word Mover’s Similarity (WMS), Sentence + Word Mover’s Similarity (S+WMS)
>
> |Dataset | Dataset Size | Text Length (Average) | Number of Annotations (Total) | Annotation Type | Annotation Procedure |
> | :------------: | :-----------: | :-----------: | :-----------: | :-----------: | :-----------: |
> | Toronto Book Corpus |  11,038 books | ~ 6,704 sentences | - | - |
> | BookCorpusOpen |  17,868 books | - | - | - | - |
> | ELTeC |  1,250 novels | - | - | - | - |
> | GLUCOSE |  4,881 stories | 5 sentences |  ~670K | Commonsense Causal Knowledge | Human-Crowdsourced (MTurk) |
> | ROCStories |  49,255 stories | 5 sentences | - | Causal & Temporal Span | Human-Crowdsourced (MTurk)  |
> | ROCStories Winter 2017 |  98,159 stories | 5 sentences | - | Causal & Temporal Span | Human-Crowdsourced (MTurk) |
> | Possible Stories | 1,313 passages  | 46.3 tokens | 8,885 ending & 4,533 questions | Alternative Ending & Causal Question  | Human-Crowdsourced (MTurk) |
> | COPA | 1K questions  | 1 sentence | 2 Alternatives for each question | the more Plausible Alternatives  | Human |
> | TIMETRAVEL |  29,849 story rewritings | 5 sentences | 81,407 counterfactual branch | Counterfactual Rewritings | Human-Crowdsourced (MTurk) |
> | HellaSwag | 70K passage  | 1 sentence | adversarial wrong answers for each passage | Counterfactual Reasoning | Machine-generated  |
> | StoryCommonsense | 15K stories | 5 sentences | 55,747 w/motiv & 104,930 w/emot| Motivations & Emotional Reactions | Human |
> | TVShowGuess | 318 characters | 137,568 tokens | 12,413 scene | Character Facts | Human-2 experts |
> | PERSONET | 33 books | 11,876 sentences |110,114 notes that contain 140,268 traits | Personalities | Automatic collection (notes from reading apps) + Human written (Crowdsourced)  |
> | LitBank | 100 fictions | 2,105.3 tokens | 29,103 mentions | Anaphoric Reference & Entity Category| Human-3 experts |
> | Phrase Detectives | 805 documents  | 1,712.4 tokens | 282,558 mentions | Anaphoric Reference | Human-Crowdsourced (game-with-a-purpose) |
> | Wanzare et al. (2019) | 504 stories | 35.74 sentences | label each sentence from 200 scenarios or none | Scenarios & Segmentation | Human-4 student assistants|
> | SNaC | 150 books | 41 sentences | 9.6K Span | Coherence Error Span & Type| Human-3 experts + Human-Crowdsourced (MTurk) |
> | Pustejovsky and Stubbs (2011) | 183 articles | - | - | Temporal Span | Human-3 students |
> | InScript | 910 stories | 12.4 sentences | 62,062 annotation for event & participant over 10 scenarios | Script Structure | Human-Crowdsourced (MTurk) |
> | Hippocorpus | 6,854 Stories |17.6 sentences | -  |Human Recalled Events | Human-Crowdsourced (MTurk) |
> | ESC v0.9 | 258 documents| - | 6,904 temporal relations & 2,265 explanatory relations | Event Relation & Temporal Span | Human-2 experts |
> | DesireDB | 3,680 instances | - | 3,680 | Fulfilment status of the desire expression sentence & Evidence span| Human-Crowdsourced (MTurk) |
> | Moral Stories | 12K stories | - |24K action + 48K consequence | Story Segment & Sentence Categories | Human-Crowdsourced (MTurk) |
> | CSI | 39 episodes (59 cases) |689 sentences per case | -  | Factual/Structural Metadata (perpetrator mentioned + relation to previous cases) | Human-3 students |
> | Shmoop | 231 stories | 112,080 tokens | 7,234 summaries for 7,234 chapters | Segmentation & chapter-level summaries  | Automatic collection (paired summaries and narrative texts from websites) |
> | BookSum | 405 books | 112,885.15 tokens | 405 summaries | Paragraph, chapter, book-level summaries  | Automatic alignment + Human inspection |
> | NARRASUM | 122K narratives | 786 tokens | 122K summaries | Book-level summaries  | Automatic alignment + Human inspection |
> | IDN-Sum | 8 IDN episodes | 3250 sentences | 10k summaries for 10k documents |  Interactive narratives summaries  | Automatic collection (paired summaries and narrative texts from websites) |
> | ABLIT | 868 chapters |154.1 sentences | 868 | Paragraph-level abridged texts  | Automatic alignment (abridged version available) + Human written (1 single author) |
> | CMU Movie Summary | 42,306 movies |176 words (median length) | 29,802 characters matched to actors with public date of birth | Character metadata  | Automatic matching |
> | Zhang et al. (2019b) | 1,036,965 stories | 15,600 words | 29,802 characters |  Character Metadata  & Tropes  | Automatic extraction + Human-100 experts (questionnaires to authors) |
> | LiSCU | 1,220 books | 1431.2 tokens (summary) | 1708 literature summaries + 9499 character descriptions | Character Description  | Automatic creation + Human evaluation (fact coverage, task difficulty)|
> | BookTest | 14,140,82 questions | 522 tokens |  141,408,250 options |  Cloze-form  | Automatic creation |
> | MCTest | 500 stories + 2000 questions | 212 words (150-300 words) | 8000 (4 questions for each story, 4 choices for each question) |  Multiple-choice  | Human-Crowdsourced (MTurk) |
> | Children’s Book Test | 108 books + 687,343 questions | 462.7 /30.7 words | 10 choices for each question | Multiple-choice | Automatic creation |
> | MovieQA | 14,944 questions | 9.3 words | 74,720 answers | Multiple-choice  | Human  |
> | WikiHow | 230,843 articles |579.8 tokens |230,843 summaries |  Subtopics & Free-form Answer  | Automatic collection |
> | MCScript2.0 | 3,487 texts + 19,821 questions |  164.4 / 8.2 tokens | 2 choices for each question |Answer Generation & Multiple-choice | Human-Crowdsourced (MTurk) |
> | Cosmos QA |35,600 paragraphs & questions | 69.4 /10.3 tokens | 4 choices for each question | Multiple-choice | Human-Crowdsourced (MTurk) |
> | NarrativeQA | 1572 documents + 46,765 questions | 61,472 / 9.8 tokens  | 46,765 answers | Free-form Answer | Human-Crowdsourced (MTurk)  |
> | TellMeWhy | 9,636 stories + 30,519 questions | 5 sentences | 3 answers for each question | Free-form Answer | Human-Crowdsourced (MTurk) |
> | FairytaleQA | 278 stories + 10,580 questions | 1401.3 / 3.3 tokens |10,580 answer & ground-truth question pairs | Free-form Answer | Human-5 postgraduate students|
>
> > 2. A task-specific example for the Bayesian estimation of the author's thoughts is necessary. The corresponding discussion was interesting but difficult to understand without being able to trace to a particular task.
>
> Thank you for the suggestion. We will provide a running example as follows:
> |Task prompt | Author prompt | Narration | BLEU-1 |
> | :------------: | :-----------: | :-----------: | :-----------: |
> | You can write plots, stories, and novels. You are an expert in tragedy and modernism. You had a life-long obsession with a girl. | The first plot is ‘Nick is invited to Gatsby’s party’. The second plot is ‘Gatsby has an affair with Daisy’. | Nick Carraway, a young man recently moved to New York City to learn about bonds, receives an unexpected invitation. It’s a lavish, handwritten note inviting him to a party at his neighbor's massive mansion. This isn't just any neighbor but Jay Gatsby, a man whose reputation for opulence precedes him… He discovers that Gatsby's ostentatious lifestyle, these grand parties, are designed to attract one particular guest - Daisy Buchanan, Nick's cousin and Gatsby's long-lost love… | 31.4 |
> | You can write plots, stories, and novels. You are an expert in fantasy fiction. You spent your childhood in Derbyshire, UK. | The first plot is ‘Nick is invited to Gatsby’s party’. The second plot is ‘Gatsby has an affair with Daisy’. | In the tranquil town of Lexington, known for its enchanting beauty and subtle charm, resides a mysterious man named Gatsby… everyone except for Nick, a humble middle-class man, who has always viewed Gatsby's world as an outsider. Nick, despite living in the same city, leads an ordinary life until one fateful day when an invitation to Gatsby's party lands on his doorstep… At the party, he meets Daisy, Gatsby's old flame, who is known for her ethereal beauty and elegance. Little does Nick know, Gatsby has been having a secret affair with Daisy, rekindling their past love. Their affair is the best-kept secret in Lexington, concealed beneath the glamour and glitz of Gatsby's extravagant lifestyle. | 27.1 |
> | You are an expert in fantasy fiction. You spent your childhood in Derbyshire, UK. You can write plots, stories, and novels.| The first plot is 'Charlie’s grandpa introduced a chocolate factory to him’. The second plot is ‘The factory owner invites Charlie to visit the factory’.|  Living in a small town of England, young Charlie Bucket's world is filled with modesty. His only escape from everyday life is through the stories told by his Grandpa Joe about an enchanting chocolate factory that once stood as the heart and soul of their town… one day the eccentric chocolatier reveals that he has decided to open up the factory once again. More incredibly, he invites Grandma Josephine to return as his assistant because of her exceptional talent and experience. However, age has taken a toll on Grandma Josephine's health. So she proposes Charlie to take her place instead… | 29.3 |
>
> Here, the narration is generated by an LLM model that simulates the author’s composition process. Task and sketch (refer to line 642) are parameterized by generative models. It could be inferred that the first line of prompts could generate the plots of The Great Gatsby and the third line of prompts correspond to Charlie and the Chocolate Factory.
>
> > 3. Line 34-37: "This is supported ... English speaker". How does this result from cognitive research support the previous statement about LLMs lacking the ability to capture the author's thoughts?
>
> This indicates a dissociation between language proficiency and thoughts in the human brain. Consequently, while LLMs might excel in adhering to linguistic rules, their competence might be limited in cognitive aspects such as logical reasoning and situation modelling.

---

### Official Review · Reviewer_xuKr · 2023-08-05

**Soundness:** 3

**Excitement:**

3: Ambivalent: It has merits (e.g., it reports state-of-the-art results, the idea is nice), but there are key weaknesses (e.g., it describes incremental work), and it can significantly benefit from another round of revision. However, I won't object to accepting it if my co-reviewers champion it.

**Missing References:**

[1] Robyn Speer, Joshua Chin, and Catherine Havasi. 2017. "ConceptNet 5.5: An Open Multilingual Graph of General Knowledge." In proceedings of AAAI 31.
[2] Baker, Collin F., Charles J. Fillmore, and John B. Lowe. "The berkeley framenet project." In COLING 1998 Volume 1: The 17th International Conference on Computational Linguistics. 1998.

**Paper Topic And Main Contributions:**

This work conducts a survey about an overview of recent narrative understanding works.The authors examine a broad categories of narrative understanding, such as taxonomy, key characteristics, datasets, evaluations, and applications. In addition, the authors propose a general framework about leveraging LLM to conduct narrative understanding, which has broad applicability in Narrative Understanding.

**Questions For The Authors:**

No detailed questions as most sections are about high-level introduction to the area. More suggestions are shared in the presentation improvement section below.

**Reasons To Accept:**

a) Survey papers dedicated to discussing narrative events are not common. The presentation can benefit researchers who are new to this area.

b) The write up is clear and easy to follow.

**Reasons To Reject:**

a) Narrative understanding could be too broad to be discussed in a 8-page conference paper. More topics and works can only be discussed with a light touch. Might consider narrowing the focus area (such as only keeping Narrative Consistency and Structural Analysis for Events), or submit as a comprehensive journal paper.

b) Lack any quantitative or qualitative studies to offer objective comparisons among works.

c) Lack of evaluations for the proposed framework.

**Reproducibility:**

N/A: Doesn't apply, since the paper does not include empirical results.

**Reviewer Confidence:**

5: Positive that my evaluation is correct. I read the paper very carefully and I am very familiar with related work.

**Typos Grammar Style And Presentation Improvements:**

a) Line 14: it is not very clear to me whether this work suggests a new method about LLM and narrative understanding. I could guess it might be related to the paragraph of Line 597. If this is one main contribution of the paper, it will be good to make this section more explicit.

b) Figure 1: the discussions in the paper do not connect to this topology very well. Suggest adding references to a specific node of the topology when discussing it in the paper.
c) Line 678: “and interactive”

d) Line 704: for world knowledge, it is crucial to include related works about Knowledge Graph, starting from early works such as ConceptNet [1], FrameNet [2], to more recent versions that connect to LLM.

---

> ### Author Rebuttal · Authors · 2023-08-29
>
> Thank you for your review. First, to address the concerns raised in your Reasons To Reject of the review:
>
> > 1. Narrative understanding could be too broad to be discussed in a 8-page conference paper. More topics and works can only be discussed with a light touch. Might consider narrowing the focus area (such as only keeping Narrative Consistency and Structural Analysis for Events), or submit as a comprehensive journal paper.
>
> We’ve seen survey papers published in conferences like EMNLP ([Ali and Hassan, 2022]( https://aclanthology.org/2022.emnlp-main.633.pdf )) and NAACL ([Wang et al., 2022]( https://aclanthology.org/2022.naacl-main.339.pdf )). These papers primarily focus on task or dataset taxonomies rather than serving as an exhaustive compendium of benchmark details and model performance. In our case, the field of narrative understanding is fast evolving, as evidenced by datasets like StoryCloze and HellaSwag for LLM evaluation ([Brown et al., 2020]( https://proceedings.neurips.cc/paper/2020/file/1457c0d6bfcb4967418bfb8ac142f64a-Paper.pdf ); [Sanh et al., 2022](https://arxiv.org/pdf/2110.08207.pdf)). Given fast development of this field, concise surveys might be better suited to track the swift shifts in dataset exploration trends. With respect to narrowing focus area, we respectfully argue that 1) "narrative understanding" is a self-contained area, with many datasets shared across various tasks. Streamlining this landscape could impede a comprehensive understanding of their overlaps and differences. 2) Present research trends often encompass a range of applications rather than focusing on a few. In our approach, we organised dispersed datasets into well-defined tasks such as reading comprehension and summarization, all consolidated within the frameworks of LLM paradigms, such as multiple choice formats and free-form text generation guided by instructions. This approach fosters both unified strategies and specialized methodologies.
>
> > 2. Lack any quantitative or qualitative studies to offer objective comparisons among works.
>
> We agree with you to make quantitative comparisons and will report model performance on representative benchmarks in the Appendix:
> |Task | Representative Dataset | Best Model | Result |
> | :------------: | :-----------: | :-----------: | :-----------: |
> | StoryCloze | StoryCloze ([Mostafazadeh et al., 2016](https://aclanthology.org/N16-1098 )) | FLAN 137B zero-shot ([Wei et al., 2021](https://openreview.net/pdf?id=gEZrGCozdqR)) | 93.4 Acc. |
> | CounterfactualReasoning | Hellaswag ([Zellers et al., 2019](https://aclanthology.org/P19-1472 )) | GPT-4 ([GPT-4 Technical Report](https://arxiv.org/abs/2303.08774 )) | 95.3 Acc.|
> |NarrativeSummarization| BookSum ([Kryscinski et al., 2022](https://aclanthology.org/2022.findings-emnlp.488.pdf )) | BART-LS ([Xiong et al., 2022](https://arxiv.org/pdf/2209.10052.pdf )) | 38.5 Rouge |
> | NarrativeQA | NarrativeQA ([Kocisky et al., 2018](https://aclanthology.org/Q18-1023 )) | Masque ([Nishida et al., 2019](https://aclanthology.org/P19-1220.pdf )) | 59.87 Rouge-L |
> | NarrativeQA | Children’s Book Test ([Hill et al., 2016](http://arxiv.org/abs/1511.02301 )) | NSE ([Dhingra et al., 2017](https://aclanthology.org/P17-1168.pdf )) | 71.9 Acc. |
>
> But because we are dedicated to offering an overview of the tasks, analyzing the performances of the models from the [leaderboards](https://paperswithcode.com/sota/sentence-completion-on-hellaswag ) will reduce the number of datasets we can cover, and is uncommon in task-oriented surveys ([Ali and Hassan, 2022]( https://aclanthology.org/2022.emnlp-main.633.pdf ); [Wang et al., 2022]( https://aclanthology.org/2022.naacl-main.339.pdf ); [Dzendzik et al., 2021]( https://aclanthology.org/2021.emnlp-main.693.pdf )). As for the qualitative comparisons, we consider Sections 2.1-2.3 as timelines of strands of works and comparisons between them when we review the developments.
>
> > 3. Lack of evaluations for the proposed framework.
>
> We will provide running examples of the author’s prompt recapitulation as soon as time and computational resources permit. But it is typically not expected that the generalized definition will be evaluated within the scope of a survey, nor are the frameworks suggested in Future Directions.
>
> We are grateful for your suggestions on presentation improvements:
> > a) Line 14: it is not very clear to me whether this work suggests a new method about LLM and narrative understanding. I could guess it might be related to the paragraph of Line 597. If this is one main contribution of the paper, it will be good to make this section more explicit.
>
> We will explain in more detail in the revised version. This work firstly aligns disparate tasks with the  LLM paradigms denoted as $p_θ(y|x, \mathsf{context})$, and categorizes  them based on the selection of $\mathsf{context}$ and the structure of $y$ (see lines 91-115, 597-612). Subsequently, it introduces Bayesian prompt selection as an alternative approach to define the task of narrative understanding (see lines 612-653).
> > b) Figure 1: the discussions in the paper do not connect to this topology very well. Suggest adding references to a specific node of the topology when discussing it in the paper. c) Line 678: “and interactive”
>
> We appreciate your review and references will be added to the nodes. The typo will be fixed as well.
> > d) Line 704: for world knowledge, it is crucial to include related works about Knowledge Graph, starting from early works such as ConceptNet [1], FrameNet [2], to more recent versions that connect to LLM.
>
> We cited LAMA ([Petroni et al., 2019](https://aclanthology.org/D19-1250.pdf )) and COMET ([Bosselut et al., 2019](https://aclanthology.org/P19-1470.pdf )) as they establish connections between ATOMIC ([Sap et al., 2019](https://ojs.aaai.org/index.php/AAAI/article/view/4160 )) and GPT, forming a foundational  static world. Additional references will be added in the revised version.

---

### Official Review · Reviewer_Bkvf · 2023-08-05

**Soundness:** 4

**Excitement:**

4: Strong: This paper deepens the understanding of some phenomenon or lowers the barriers to an existing research direction.

**Missing References:**

I suggest adding this dataset to Table A1:
Ladhak, Faisal, et al. "Exploring Content Selection in Summarization of Novel Chapters." Proceedings of the 58th Annual Meeting of the Association for Computational Linguistics. 2020.

**Paper Topic And Main Contributions:**

This paper conducts a comprehensive survey and taxonomy of narrative understanding tasks, including reading comprehension, summarization, and question answering, along with their associated datasets and evaluation measures. The final section explores the potential of extending the capabilities of modularized LLMs to tackle novel narrative understanding tasks.

**Questions For The Authors:**

A. In Fig1, should the citation for narrative summarization->event be zhao 2022a instead of b?

B. When building the task taxonomy, why is reading comprehension considered to have categorical labels while question answering is treated as an isolated branch instead of a reading comprehension task?

C. Should some narrative-specific NLP tasks such as novel NER also be considered understanding tasks or do you prefer to regard them as low-level processing tasks?

**Reasons To Accept:**

This paper presents a comprehensive survey and taxonomy of narrative understanding tasks.

**Reasons To Reject:**

I didn't find significant weaknesses. While it would be nice to include narrative generation in the survey, I understand that it may exceed the scope of this submission. For future directions, it would be valuable to have more discussions about the intrinsic challenges of narrative understanding, not necessarily limited to the capabilities of LLMs.

**Reproducibility:**

N/A: Doesn't apply, since the paper does not include empirical results.

**Reviewer Confidence:**

4: Quite sure. I tried to check the important points carefully. It's unlikely, though conceivable, that I missed something that should affect my ratings.

---

> ### Author Rebuttal · Authors · 2023-08-29
>
> Thank you for your review! To address your specific concerns:
> > 1.  For future directions, it would be valuable to have more discussions about the intrinsic challenges of narrative understanding, not necessarily limited to the capabilities of LLMs.
>
> Narrative understanding presents many challenges due to the complex nature of human storytelling. Some of the challenges include: (1) Character identification and event tracking.  Character identification needs to deal with the challenges of co-referencing and entity linking, while unsupervised event detection approaches need to be developed to recognise main events with an arbitrary set of arguments from text. (2) Dynamic character network construction. The relationship between characters may not be static, but evolving over time. Also the relation types cannot be pre-defined. (3) Plot structure analysis. Traditionally, an event storyline usually organises related events in a chronological order. However, in narratives, stories are complex and the narrative could be structured in many different ways including flashbacks, multiple stories happening simultaneously, many smaller stories taking place within a framework of a bigger plot, or even intertwining stories. Developing methods to automatically identify plot structured elements is challenging. More discussions will be added in the revised version,
>
> > 2. In Fig1, should the citation for narrative summarization->event be zhao 2022a instead of b?
>
> Yes, the correct citation should be 'Zhao 2022a' instead of 'Zhao 2022b.' The '2022a' paper focuses on events, while the '2022b' paper is centred on NarrativeQA.
>
> > 3. When building the task taxonomy, why is reading comprehension considered to have categorical labels while question answering is treated as an isolated branch instead of a reading comprehension task?
>
> This is precisely the issue we considered when building the task taxonomy: Question answering is one of the most popular ways for evaluating reading comprehension tasks. Reading comprehension takes the form of QA in datasets like NarrativeQA, TellMeWhy, and FairytaleQA. With the advancement of Large Language Models (LLMs), the scope of generative-based question answering has significantly widened. Depending on the specific questions posed, its scope can encompass other more fine-grained tasks defined in our taxonomy, including narrative summarization (Who is Charlie?), and narrative generation (What would happen after the Princess marries the Prince?). Therefore, we decided to treat it as an isolated branch.
>
> > 4. Should some narrative-specific NLP tasks such as novel NER also be considered understanding tasks or do you prefer to regard them as low-level processing tasks?
>
> Some narrative-specific NLP tasks such as Named Entity Recognition (NER) might be perceived as understanding-related tasks when determining whether designating 'a talking cup' as a 'person' is appropriate. We will add a section on 'future work' to discuss the modification needed for traditional NLP tasks to be effectively applied to narratives.
>
> > 5. I suggest adding this dataset to Table A1: Ladhak, Faisal, et al. "Exploring Content Selection in Summarization of Novel Chapters." Proceedings of the 58th Annual Meeting of the Association for Computational Linguistics. 2020.
>
> Thanks for the suggestion, we will add this dataset to Table A1.
>
> |Dataset | Reference | Task | Language | Domain | Evaluation |
> | :------------: | :-----------: | :-----------: | :-----------: | :-----------: | :-----------: |
> | NovelChapter| [Ladhak et al., 2021](https://aclanthology.org/2020.acl-main.453.pdf)  | Corpus-Level Summarisation | English | Novel | Similarity Metrics |
>
> |Dataset | Dataset Size | Text Length (Average) | Number of Annotations (Total) | Annotation Type | Annotation Procedure |
> | :------------: | :-----------: | :-----------: | :-----------: | :-----------: | :-----------: |
> | NovelChapter |  4,383 chapters | 5,165 words | 8,088 chapter/summary pairs | Chapter-level summaries | Automatic collection (summary-chapter pairs online study guides) + Human written (expert) |

---

### Meta-Review · Area_Chair_G9Pg · 2023-09-15

**Recommendation:** 4

**Metareview:**

The reviewers collectively see merit in this comprehensive survey, both in terms of the content and style of the paper. The main issues seem to be related to the 8-page length restriction. Because of this restriction, more detailed descriptions, analyses, and examples are lacking, which may be added if the paper is revised as a journal article.

Paper Topic And Main Contributions:

This paper provides a comprehensive survey of narrative understanding tasks, encompassing reading comprehension, summarization, and question answering. It explores various aspects of narrative understanding, including taxonomy, datasets, evaluation measures, and potential applications. Additionally, the authors propose a novel framework leveraging Large Language Models (LLMs) to tackle narrative understanding tasks, emphasizing the estimation of latent author thoughts using task descriptions and prompts.

Reasons To Accept:

* Comprehensive Survey, Useful Resource, and Insights for Future Directions
    * The paper provides an in-depth and comprehensive survey of narrative understanding tasks, making it a valuable resource for both new and experienced researchers in the field.
    * The paper serves as a useful resource for researchers by providing an index of narrative understanding datasets, facilitating easy access to essential research materials in the domain.
    * The paper offers valuable insights into the future directions of narrative understanding tasks, including the proposal of a Bayesian estimation approach for understanding the author's thoughts. This forward-looking perspective adds significant value to the paper.
* Clarity and Accessibility
    * The write-up is clear and easy to follow, enhancing its accessibility to a wide range of readers.

Reasons To Reject:
* Maybe Better Suited as a Journal Article:
    * Because the paper attempts to cover a broad range of narrative understanding topics, this exceeds the depth that can be provided in an 8-page conference paper. This may suggest the need for a narrower focus or a submission as a comprehensive journal paper.
* Issues with the Proposed Framework:
    * The paper introduces a framework for narrative understanding but does not provide any evaluations or practical implementations of this framework, which leaves its effectiveness unproven.
    * The authors could have strengthened the paper by providing a more fine-grained categorization of tasks and datasets, considering factors such as dataset size, annotation procedure, domain, period, language, etc., which would have added depth to the survey.
* Need for More Analyses and Examples:
    * The paper lacks quantitative and qualitative studies that would allow for objective comparisons among the works discussed. This absence of empirical evaluation or analysis reduces the paper's contribution.
    * While the paper discusses the Bayesian estimation of the author's thoughts, it lacks a concrete task-specific example, making it challenging for readers to understand how this concept can be applied in practice.

---

### Decision · Program_Chairs · 2023-10-07

**Decision:**

Accept-Findings

**Comment:**

The reviewers collectively see merit in this comprehensive survey, both in terms of the content and style of the paper. The main issues seem to be related to the 8-page length restriction. Because of this restriction, more detailed descriptions, analyses, and examples are lacking, which may be added if the paper is revised as a journal article.

Paper Topic And Main Contributions:

This paper provides a comprehensive survey of narrative understanding tasks, encompassing reading comprehension, summarization, and question answering. It explores various aspects of narrative understanding, including taxonomy, datasets, evaluation measures, and potential applications. Additionally, the authors propose a novel framework leveraging Large Language Models (LLMs) to tackle narrative understanding tasks, emphasizing the estimation of latent author thoughts using task descriptions and prompts.

Reasons To Accept:

* Comprehensive Survey, Useful Resource, and Insights for Future Directions
    * The paper provides an in-depth and comprehensive survey of narrative understanding tasks, making it a valuable resource for both new and experienced researchers in the field.
    * The paper serves as a useful resource for researchers by providing an index of narrative understanding datasets, facilitating easy access to essential research materials in the domain.
    * The paper offers valuable insights into the future directions of narrative understanding tasks, including the proposal of a Bayesian estimation approach for understanding the author's thoughts. This forward-looking perspective adds significant value to the paper.
* Clarity and Accessibility
    * The write-up is clear and easy to follow, enhancing its accessibility to a wide range of readers.

Reasons To Reject:
* Maybe Better Suited as a Journal Article:
    * Because the paper attempts to cover a broad range of narrative understanding topics, this exceeds the depth that can be provided in an 8-page conference paper. This may suggest the need for a narrower focus or a submission as a comprehensive journal paper.
* Issues with the Proposed Framework:
    * The paper introduces a framework for narrative understanding but does not provide any evaluations or practical implementations of this framework, which leaves its effectiveness unproven.
    * The authors could have strengthened the paper by providing a more fine-grained categorization of tasks and datasets, considering factors such as dataset size, annotation procedure, domain, period, language, etc., which would have added depth to the survey.
* Need for More Analyses and Examples:
    * The paper lacks quantitative and qualitative studies that would allow for objective comparisons among the works discussed. This absence of empirical evaluation or analysis reduces the paper's contribution.
    * While the paper discusses the Bayesian estimation of the author's thoughts, it lacks a concrete task-specific example, making it challenging for readers to understand how this concept can be applied in practice.